# Monte-Carlo Tree Search for Constrained POMDPs

**Jongmin Lee[1], Geon-Hyeong Kim[1], Pascal Poupart[2], Kee-Eung Kim[1,3]**
[1] School of Computing, KAIST, Republic of Korea
[2] University of Waterloo, Waterloo AI Institute and Vector Institute
[3] PROWLER.io
{jmlee,ghkim}@ai.kaist.ac.kr, ppoupart@uwaterloo.ca, kekim@cs.kaist.ac.kr

## Abstract

Monte-Carlo Tree Search (MCTS) has been successfully applied to very large POMDPs, a standard model for stochastic sequential decision-making problems. However, many real-world problems inherently have multiple goals, where multi-objective formulations are more natural. The constrained POMDP (CPOMDP) is such a model that maximizes the reward while constraining the cost, extending the standard POMDP model. To date, solution methods for CPOMDPs assume an explicit model of the environment, and thus are hardly applicable to large-scale real-world problems. In this paper, we present CC-POMCP (Cost-Constrained POMCP), an online MCTS algorithm for large CPOMDPs that leverages the optimization of LP-induced parameters and only requires a black-box simulator of the environment. In the experiments, we demonstrate that CC-POMCP converges to the optimal stochastic action selection in CPOMDP and pushes the state-of-the-art by being able to scale to very large problems.

## 1 Introduction

Monte-Carlo Tree Search (MCTS) [4, 5, 12] is a generic online planning algorithm that effectively combines random sampling and tree search, and has shown great promise in many areas such as online Bayesian reinforcement learning [8, 10] and computer Go [7, 20]. MCTS efficiently explores the search space by investing more search effort in promising states and actions while balancing exploration and exploitation in the direction of maximizing the cumulative (scalar) rewards. Due to its outstanding performance without relying on any prior domain knowledge or heuristic function, MCTS has become the de-facto standard method for solving very large sequential decision making problems, commonly formulated as Markov decision processes (MDPs) and partially observable MDPs (POMDPs).

However in many situations, it is not straightforward to formulate the objective with the reward maximization alone, as in the following examples. For spoken dialogue systems [24], it is common to optimize the dialogue strategy towards minimizing the number of turns while maintaining the success rate of dialogue tasks above a certain level. For UAVs under search and rescue mission, the main goal would be to find as many targets as possible, while avoiding threats that may endanger the mission itself. The constrained POMDP (CPOMDP) [9] is an appealing framework for dealing with this kind of multi-objective sequential decision making problems when the environment is partially observable. The model assumes that the action incurs not only rewards, but also $K$ different types of costs, and the goal is to find an optimal policy that maximizes the expected cumulative rewards while bounding each of $K$ expected cumulative costs below certain levels.

Although the CPOMDP is a very expressive model, it is known to be very difficult to solve due to the PSPACE-complete nature of solving POMDP [16] originating from the two main challenges: the *curse of dimensionality* and the *curse of history*. Partially observable Monte-Carlo Planner (POMCP) [19] tames these two curses of POMDP by using Monte-Carlo sampling both in the root belief-state

and in the black-box simulation of the history. In contrast, solution methods for CPOMDPs, e.g. dynamic programming [11], linear programming [18], and online uniform-cost search [23], are not yet advanced up to this level, partly due to requiring an explicit model of the environment. This prevents CPOMDP from being a practical approach for modeling real-world applications.

In this paper, we present an MCTS algorithm for CPOMDPs, which precisely addresses the scalability. To the best of our knowledge, extending MCTS to CPOMDPs (or even CMDPs) has remained unexplored since it is not straightforward to handle the constrained optimization in tree search. This challenge is compounded by the fact that optimal policies can be stochastic.[1] In order to develop MCTS for CPOMDPs, we first show that solving CPOMDPs is essentially equivalent to jointly solving an unconstrained POMDP while optimizing its LP-induced parameters that control the trade-off between the reward and the costs. From this result, we present our algorithm, Cost-Constrained POMCP (CC-POMCP), for solving large CPOMDPs that combine traditional MCTS with LP-induced parameter optimization. In the experiments section, we demonstrate that CC-POMCP converges to the optimal stochastic action selection using a synthetic domain and that it is able to handle very large problems including constrained variants of Rocksample(15,15) and Atari 2600 arcade game, pushing the state-of-the-art scalability in CPOMDP solvers.

## 2 Background

Partially observable Markov decision processes (POMDPs) [22] provide a principled framework for modeling sequential decision making problems under stochastic transitions and noisy observations. It is formally defined by tuple $\langle S_p, A, O_p, T_p, Z_p, R_p, \gamma, b_0 \rangle$, where $S_p$ is the set of environment states $s$, $A$ is the set of actions $a$, $O_p$ is the set of observations $o$, $T_p(s'|s,a) = \Pr(s_{t+1} = s'|s_t = s, a_t = a)$ is the transition probability, $Z_p(o|s',a) = \Pr(o_{t+1} = o|s_{t+1} = s', a_t = a)$ is the observation probability, $R_p(s,a) \in \mathbb{R}$ is the immediate reward for taking action $a$ in state $s$, $\gamma \in [0,1)$ is the discount factor, and $b_0(s) = \Pr(s_0 = s)$ is the starting state distribution at time step 0. The history $h_t = [a_0, o_0, \ldots, a_t, o_t]$ and $h_t a_{t+1} = [a_0, o_0, \ldots, a_t, o_t, a_{t+1}]$ denote the sequence of actions and observations. The agent takes an action via policy $\pi(a|h) = \Pr(a_t = a|h_t = h)$ that maps from history to probability distribution over actions. In POMDPs, the environment state is not directly observable, thus the agent maintains belief $b_t(s) = \Pr(s_t = s|h_t)$ that can be recursively updated using the Bayes rule: when taking action $a$ in $b$ and observing $o$, the updated belief is $b^{ao}(s') \propto Z_p(o|s',a) \sum_s T_p(s'|s,a)b(s)$. Since belief $b_t$ is a sufficient statistic of history $h_t$, the POMDP can be understood as the belief-state MDP $\langle B, A, T, R, \gamma, b_0 \rangle$, where $b_0$ is the initial state, $B$ is the set of reachable beliefs starting from $b_0$, $T(b'|b,a) = \sum_{o,s,s'} Z_p(o|s',a)T_p(s'|s,a)b(s)\delta(b', b^{ao})$ is the transition probability, $R(b,a) = \sum_s b(s)R_p(s,a)$ is the immediate reward function. We shall use $h$ and $b = \Pr(s|h)$ interchangeably as long as there is no confusion (e.g. $Q_R(h,a) = Q_R(b,a)$, $\pi(a|h) = \pi(a|b)$). The goal is to find an optimal policy $\pi^*$ that maximizes the expected discounted return (i.e. cumulative discounted rewards):

$$\max_\pi V_R^\pi(b_0) = E_\pi \left[ \sum_{t=0}^\infty \gamma^t R(b_t, a_t) \,\Big|\, b_0 \right].$$

Constrained POMDPs (CPOMDPs) [9, 11, 18] is a generalization of POMDPs for multi-objective problems. It is formally defined by tuple $\langle S_p, A, O_p, T_p, Z_p, R_p, \mathbf{C}_p, \hat{\mathbf{c}}, \gamma, b_0 \rangle$, where $\mathbf{C}_p = \{C_{p,k}\}_{1..K}$ is the set of $K$ non-negative cost functions with individual thresholds $\hat{\mathbf{c}} = \{\hat{c}_k\}_{1..K}$. Similarly, a CPOMDP can be cast into an equivalent belief-state CMDP $\langle B, A, T, R, \mathbf{C}, \hat{\mathbf{c}}, \gamma \rangle$, where $C_k(b,a) = \sum_s b(s)C_{p,k}(s,a)$. The goal is to compute an optimal policy that maximizes the expected cumulative reward while bounding the expected cumulative costs:

$$\max_\pi V_R^\pi(b_0) = \mathbb{E}_\pi \left[ \sum_{t=0}^\infty \gamma^t R(b_t, a_t) \,\Big|\, b_0 \right]$$

$$s.t. \ V_{C_k}^\pi(b_0) = \mathbb{E}_\pi \left[ \sum_{t=0}^\infty \gamma^t C_k(b_t, a_t) \,\Big|\, b_0 \right] \le \hat{c}_k \ \ \forall k$$

An optimal policy of the CPOMDP (or the equivalent belief-state CMDP) is generally stochastic and can be obtained by solving the following linear program (LP) [1]:

$$\max_{\{y(b,a) \geq 0\} \forall b,a} \sum_{b,a} R(b,a)y(b,a) \tag{1}$$

$$s.t. \sum_{a'} y(b',a') = \delta(b_0, b') + \gamma \sum_{b,a} T(b'|b,a)y(b,a) \quad \forall b'$$

$$\sum_{b,a} C_k(b,a)y(b,a) \leq \hat{c}_k \quad \forall k$$

where $y(b,a)$ can be interpreted as a discounted occupancy measure of $(b,a)$, and $\delta(x,y)$ is a Dirac delta function that has the value of 1 if $x = y$ and 0 otherwise. Once the optimal solution $y^*(b,a)$ is obtained, an optimal stochastic policy and the corresponding optimal value are computed by $\pi^*(a|b) = \Pr(a|b) = y^*(b,a)/\sum_{a'} y^*(b,a')$ and $V_R^*(b_0; \hat{\mathbf{c}}) = \sum_{b,a} R(b,a)y^*(b,a)$ respectively. It is usually intractable to solve LP (1) exactly since the cardinality of $B$ can be infinite.

POMCP [19] is a highly scalable Monte-Carlo tree search (MCTS) algorithm for (unconstrained) POMDPs. The algorithm uses Monte-Carlo simulation for both tree search and belief update to simultaneously tackle the *curse of history* and the *curse of dimensionality*. In each simulation, a state particle is sampled from the root node's belief-state $s \sim \mathcal{B}(h)$ (called root sampling) and is used to sample a trajectory using a black-box simulator $(s', o, r) \sim \mathcal{G}(s, a)$. It adopts UCB1 [2] as the tree policy, i.e. the action selection rule in the internal nodes of the search tree:

$$\arg\max_a \left[ Q_R(h,a) + \kappa \sqrt{\frac{\log N(h)}{N(h,a)}} \right]$$

where $Q_R(h,a)$ is the average of the sampled returns, $N(h)$ is the number of simulations performed through $h$, $N(h,a)$ is the number of times action $a$ is selected in $h$, and $\kappa$ is the exploration constant to adjust the exploration-exploitation trade-off. POMCP expands the search tree non-uniformly, focusing more search efforts in promising nodes. It can be formally shown that $Q_R(h,a)$ asymptotically converges to the optimal value $Q_R^*(h,a)$ in POMDPs.

Unfortunately, it is not straightforward to use POMCP for CPOMDPs since the original UCB1 action selection rule does not have any notion of cost constraints. If we naively adopt the vanilla, reward-maximizing POMCP, we may obtain cost-violating action sequences. We could obtain the average of sampled cumulative costs $Q_C$ during search, but it is not straightforward how to leverage them in the tree policy: if we naively prevent action branches that violate the cost constraint $Q_C(h,a) \leq \hat{c}$, we may end up with policies that are too conservative and thus sub-optimal, i.e. a feasible policy may be rejected during search if the Monte-Carlo estimate violates the cost constraint.

## 3   Solving CPOMDP via a POMDP Solver

The derivation of our algorithm starts from the dual of (1):

$$\min_{\substack{\{\mathcal{V}(b)\} \forall b \\ \{\lambda_k \geq 0\} \forall k}} \sum_b \delta(b_0, b)\mathcal{V}(b) + \sum_k \hat{c}_k \lambda_k \tag{2}$$

$$s.t. \quad \mathcal{V}(b) \geq R(b,a) - \sum_k C_k(b,a)\lambda_k + \gamma \sum_{b'} T(b'|b,a)\mathcal{V}(b') \quad \forall b,a$$

Observe that if we treat $\boldsymbol{\lambda} = [\lambda_1, \ldots, \lambda_K]^\top$ as a constant, the problem becomes an *unconstrained* belief-state MDP with the scalarized reward function $R(b,a) - \boldsymbol{\lambda}^\top \mathbf{C}(b,a)$. Let $\mathcal{V}_{\boldsymbol{\lambda}}^*$ be the optimal value function of this unconstrained POMDP. Then, for any $\boldsymbol{\lambda}$, there exists a corresponding unique $\mathcal{V}_{\boldsymbol{\lambda}}^*$, and we can compute $\mathcal{V}_{\boldsymbol{\lambda}}^*$ with a POMDP solver. Thus, solving the dual LP (2) reduces to:

$$\min_{\boldsymbol{\lambda} \geq \mathbf{0}} \left[ \mathcal{V}_{\boldsymbol{\lambda}}^*(b_0) + \boldsymbol{\lambda}^\top \hat{\mathbf{c}} \right] \tag{3}$$

Moreover, if there is an optimal solution $y^*$ to the primal LP in (1), then there exists a corresponding dual optimal solution $\mathcal{V}^*$ and $\boldsymbol{\lambda}^*$, and the duality gap is zero, i.e.

$$V_R^*(b_0; \hat{\mathbf{c}}) = \sum_{b,a} R(b,a)y^*(b,a) = \mathcal{V}_{\boldsymbol{\lambda}^*}^*(b_0) + \boldsymbol{\lambda}^{*\top} \hat{\mathbf{c}}$$

by the strong duality theorem.

To compute optimal $\boldsymbol{\lambda}$ in Eq. (3), we have to consider the trade-off between the first term and the second term according to the cost constraint $\hat{\mathbf{c}}$. For example, if the cost constraint $\hat{\mathbf{c}}$ is very large, the optimal solution $\boldsymbol{\lambda}^*$ tends to be close to zero since the objective function would be mostly affected by the second term $\boldsymbol{\lambda}^\top \hat{\mathbf{c}}$. On the other hand, if $\hat{\mathbf{c}}$ is sufficiently small, the first term will be dominant and the optimal solution $\boldsymbol{\lambda}^*$ tends to get larger in order to have a negative impact on the reward $R(b, a) - \boldsymbol{\lambda}^\top \mathbf{C}(b, a)$. Thus, it may seem that Eq. (3) is a complex optimization problem. However, as we will see in the following proposition, the objective function in Eq. (3) is actually piecewise-linear and convex over $\boldsymbol{\lambda}$, as depicted in Figure 3 in Appendix A.

**Proposition 1.** *Let $\mathcal{V}_{\boldsymbol{\lambda}}^*$ be the optimal value function of the POMDP with scalarized reward function $R(b, a) - \boldsymbol{\lambda}^\top \mathbf{C}(b, a)$. Then, $\mathcal{V}_{\boldsymbol{\lambda}}^*(b_0) + \boldsymbol{\lambda}^\top \hat{\mathbf{c}}$ is a piecewise-linear and convex (PWLC) function of $\boldsymbol{\lambda}$. (The proof is provided in Appendix A.)*

In addition, we can show that the optimal solution $\boldsymbol{\lambda}^*$ is bounded:

**Proposition 2** (Lemma 4 in [14]). *Suppose that the reward function is bounded in $[R_{\min}, R_{\max}]$ and there exists $\tau > 0$ and a (feasible) policy $\pi$ such that $V_{\mathbf{C}}^\pi(b_0) + \tau \mathbf{1} \leq \hat{\mathbf{c}}$. Then, $\|\boldsymbol{\lambda}^*\|_1 \leq \frac{R_{\max} - R_{\min}}{\tau(1-\gamma)}$.*

Thus, from Propositions 1 and 2, we can obtain optimal $\boldsymbol{\lambda}^*$ by greedily optimizing (3) with $\lambda_k$ in the range $[0, \frac{R_{\max} - R_{\min}}{\tau(1-\gamma)}]$. The remaining question is how to compute that direction for updating $\boldsymbol{\lambda}$. We start with the following lemma to answer this question:

**Lemma 1.** *Let $\mathcal{M}_1 = \langle B, A, T, R_1, \gamma \rangle$ and $\mathcal{M}_2 = \langle B, A, T, R_2, \gamma \rangle$ be two (belief-state) MDPs differing only in the reward function, and $V_1^\pi$ and $V_2^\pi$ be the value functions of $\mathcal{M}_1$ and $\mathcal{M}_2$ with a fixed policy $\pi$. Then, the value function of the new MDP $\mathcal{M} = \langle B, A, T, pR_1 + qR_2, \gamma \rangle$ with the policy $\pi$ is $V^\pi(b) = pV_1^\pi(b) + qV_2^\pi(b)$ for all $b \in B$. (The proof is provided in Appendix B.)*

Lemma 1 implies that $\mathcal{V}_{\boldsymbol{\lambda}}^*$ can be decomposed into $\mathcal{V}_{\boldsymbol{\lambda}}^*(b_0) = V_R^{\pi_{\boldsymbol{\lambda}}^*}(b_0) - \boldsymbol{\lambda}^\top V_{\mathbf{C}}^{\pi_{\boldsymbol{\lambda}}^*}(b_0)$ where $\pi_{\boldsymbol{\lambda}}^*$ is the optimal policy with respect to the scalarized reward function $R(b, a) - \boldsymbol{\lambda}^\top \mathbf{C}(b, a)$, and thus (3) becomes:

$$\min_{\boldsymbol{\lambda} \geq \mathbf{0}} \left[ V_R^{\pi_{\boldsymbol{\lambda}}^*}(b_0) - \boldsymbol{\lambda}^\top V_{\mathbf{C}}^{\pi_{\boldsymbol{\lambda}}^*}(b_0) + \boldsymbol{\lambda}^\top \hat{\mathbf{c}} \right] \tag{4}$$

One way to compute the descent direction for $\boldsymbol{\lambda}$ would be by taking the derivative of Eq. (4) with respect to $\boldsymbol{\lambda}$ while holding $\pi_{\boldsymbol{\lambda}}^*$ constant so that we use the direction $V_{\mathbf{C}}^{\pi_{\boldsymbol{\lambda}}^*}(b_0) - \hat{\mathbf{c}}$. The following theorem shows that this is indeed a valid direction:

**Theorem 1.** *For any $\boldsymbol{\lambda}$, $V_{\mathbf{C}}^{\pi_{\boldsymbol{\lambda}}^*}(b_0) - \hat{\mathbf{c}}$ is a negative subgradient that decreases the objective in Eq. (3), where $\pi_{\boldsymbol{\lambda}}^*$ is the optimal policy with respect to the scalarized reward function $R(b, a) - \boldsymbol{\lambda}^\top \mathbf{C}(b, a)$. Also, if $V_{\mathbf{C}}^{\pi_{\boldsymbol{\lambda}}^*}(b_0) - \hat{\mathbf{c}} = \mathbf{0}$ then $\boldsymbol{\lambda}$ is the optimal solution of Eq. (3). (Proof provided in Appendix C.)*

The direction $V_{\mathbf{C}}^{\pi_{\boldsymbol{\lambda}}^*}(b_0) - \hat{\mathbf{c}}$ has a natural interpretation: if the current policy violates the $k$-th cost constraint (i.e. $V_{C_k}^{\pi_{\boldsymbol{\lambda}}^*} > \hat{c}_k$), $\lambda_k$ increases so that the cost is penalized more in the scalarized reward function $R(b, a) - \boldsymbol{\lambda}^\top \mathbf{C}(b, a)$. On the other hand, if the current policy is too conservative for the $k$-th cost constraint (i.e. $V_{C_k}^{\pi_{\boldsymbol{\lambda}}^*} < \hat{c}_k$), $\lambda_k$ decreases so that the cost is penalized less.

In summary, we can solve the dual of LP of the belief-state CMDP by iterating through the following steps, starting from any $\boldsymbol{\lambda}$:

1. $\pi_{\boldsymbol{\lambda}}^* \leftarrow \text{SolveBeliefMDP}(\langle B, A, T, R - \boldsymbol{\lambda}^\top \mathbf{C}, \gamma \rangle)$

2. $V_{\mathbf{C}}^{\pi_{\boldsymbol{\lambda}}^*} \leftarrow \text{PolicyEvaluation}(\langle B, A, T, \mathbf{C}, \gamma \rangle, \pi_{\boldsymbol{\lambda}}^*)$

3. $\boldsymbol{\lambda} \leftarrow \boldsymbol{\lambda} + \alpha_n (V_{\mathbf{C}}^{\pi_{\boldsymbol{\lambda}}^*}(b_0) - \hat{\mathbf{c}})$ and clip $\lambda_k$ to range $[0, \frac{R_{\max} - R_{\min}}{\tau(1-\gamma)}] \quad \forall k \in \{1, 2, ...K\}$

By Theorem 1, this procedure is a subgradient method, guaranteed to converge to the optimal solution by using a step-size sequence $\alpha_n$ such that $\sum_n \alpha_n = \infty$ and $\sum_n \alpha_n^2 < \infty$.

**Algorithm 1** Cost-Constrained POMCP (CC-POMCP)

**function** SEARCH($h_0$)
    $\lambda$ is randomly initialized.
    **repeat**
        **if** $h = \emptyset$ **then**
            $s \sim b_0$
        **else**
            $s \sim \mathcal{B}(h_0)$
        **end if**
        SIMULATE($s, h_0, 0$)
        $a \sim$ GREEDYPOLICY($h_0, 0, 0$)
        $\lambda \leftarrow \lambda + \alpha_n \left[Q_C(h_0, a) - \hat{c}\right]$
        Clip $\lambda_k$ to range $[0, \frac{R_{\max} - R_{\min}}{\tau(1-\gamma)}]$  $\forall k = \{1, 2, ... K\}$
    **until** TIMEOUT()
    **return** GREEDYPOLICY($h_0, 0, \nu$)
**end function**

**function** ROLLOUT($s, h, d$)
    **if** $d =$ (maximum-depth) **then**
        **return** $[0, 0]$
    **end if**
    $a \sim \pi_{rollout}(\cdot|h)$ and $(s', o, r, c) \sim \mathcal{G}(s, a)$
    **return** $[r, c] + \gamma \cdot$ ROLLOUT($s', hao, d+1$)
**end function**

**function** GREEDYPOLICY($h, \kappa, \nu$)
    $\mathcal{Q}_\lambda^\oplus(h, a) := Q_R(h, a) - \lambda^\top Q_C(h, a) + \kappa\sqrt{\frac{\log N(h)}{N(h,a)}}$
    $a^* \leftarrow \arg\max_a \mathcal{Q}_\lambda^\oplus(h, a)$
    $A^* \leftarrow \Big\{ a_i^* \mid |\mathcal{Q}_\lambda(h, a_i^*) - \mathcal{Q}_\lambda(h, a^*)|$
           $\leq \nu\Big(\sqrt{\frac{\log N(h, a_i^*)}{N(h, a_i^*)}} + \sqrt{\frac{\log N(h, a^*)}{N(h, a^*)}}\Big)\Big\}$
    Solve LP (10) with $A^*$ to compute a policy $\pi(a_i^*|h) = w_i$.

    **return** $\pi(\cdot|h)$
**end function**

**function** SIMULATE($s, h, d$)
    **if** $d =$ (maximum-depth) **then**
        **return** $[0, 0]$
    **end if**
    **if** $h \notin T$ **then**
        $T(ha) \leftarrow (N_{init}, Q_{R,init}, Q_{C,init}, \emptyset) \; \forall a$
        **return** ROLLOUT($s, h, d$)
    **end if**
    $a \sim$ GREEDYPOLICY($h, \kappa, \nu$)
    $(s', o, r, c) \sim \mathcal{G}(s, a)$
    $[R, C] \leftarrow [r, c] + \gamma \cdot$ SIMULATE($s', hao, d+1$)
    $\mathcal{B}(h) \leftarrow \mathcal{B}(h) \cup \{s\}$
    $N(h) \leftarrow N(h) + 1$
    $N(h, a) \leftarrow N(h, a) + 1$
    $Q_R(h, a) \leftarrow Q_R(h, a) + \frac{R - Q_R(h,a)}{N(h,a)}$
    $Q_C(h, a) \leftarrow Q_C(h, a) + \frac{C - Q_C(h,a)}{N(h,a)}$
    $\bar{c}(h, a) \leftarrow \bar{c}(h, a) + \frac{c - \bar{c}(h,a)}{N(h,a)}$
    **return** $[R, C]$
**end function**

**function** MAINLOOP()
    $\hat{c} \leftarrow$ (cost constraint)
    $s \leftarrow$ (initial state)
    $h \leftarrow \emptyset$
    **while** $s$ is not terminal **do**
        $\pi \leftarrow$ SEARCH($h$)
        $a \sim \pi(\cdot|h)$
        $(s', o, r, c) \sim \mathcal{G}(s, a)$
        $\hat{c} \leftarrow \frac{\hat{c} - \pi(a|h)\bar{c}(h,a) - \sum_{a' \neq a} \pi(a'|h)Q_C(h,a')}{\gamma\pi(a|h)}$
        $s \leftarrow s'$
        $h \leftarrow hao$
    **end while**
**end function**

## 4   Cost-Constrained POMCP (CC-POMCP)

Although we have eliminated the cost-constraints by introducing simultaneous update of $\boldsymbol{\lambda}$, it still relies on exactly solving POMDPs via *SolveBeliefMDP* in each iteration, which is impractical for large CPOMDPs. Fortunately, all we need in step 3 is the cost value at the *initial* belief state $V_{\mathbf{C}}^{\pi_\lambda^*}(b_0)$ with respect to the optimal policy when the reward function is given by $R - \boldsymbol{\lambda}^\top \mathbf{C}$. This is exactly the situation where MCTS can be effectively applied: MCTS focuses on finding the optimal action selection at the *root node* using the Monte-Carlo estimate of long-term rewards (or costs). We are now ready to present our online algorithm for large CPOMDPs, which we refer to as Cost-Constrained POMCP (CC-POMCP), shown in Algorithm 1. The changes from the standard POMCP are highlighted in blue. CC-POMCP is an extension of POMCP with cost constraints and can be seen as an anytime approximation of policy iteration with the simultaneous optimization of $\boldsymbol{\lambda}$: the policy is sequentially evaluated via Monte-Carlo return

$$Q_R(h, a) \leftarrow Q_R(h, a) + \frac{R - Q_R(h, a)}{N(h, a)} \quad \text{and} \quad Q_{\mathbf{C}}(h, a) \leftarrow Q_{\mathbf{C}}(h, a) + \frac{\mathbf{C} - Q_{\mathbf{C}}(h, a)}{N(h, a)} \quad (5)$$

and the policy is implicitly improved by the UCB1 action selection rule based on the scalarized value $\mathcal{Q}_{\boldsymbol{\lambda}}(h, a) = Q_R(h, a) - \boldsymbol{\lambda}^\top Q_{\mathbf{C}}(h, a)$:

$$\arg\max_a \mathcal{Q}_{\boldsymbol{\lambda}}^\oplus(h, a) = \left[ Q_R(h, a) - \boldsymbol{\lambda}^\top Q_{\mathbf{C}}(h, a) + \kappa\sqrt{\frac{\log N(h)}{N(h, a)}} \right] \quad (6)$$

Finally, $\boldsymbol{\lambda}$ is updated simultaneously using the current estimate of $V_{\mathbf{C}}(s_0) - \hat{\mathbf{c}}$, which is the descent direction of the convex objective function:

$$\boldsymbol{\lambda} \leftarrow \boldsymbol{\lambda} + \alpha_n(Q_{\mathbf{C}}(h_0, a) - \hat{\mathbf{c}}) \text{ where } a \sim \pi(\cdot|h_0) \tag{7}$$

The following theorem states that CC-POMCP asymptotically converges to optimal $\boldsymbol{\lambda}^*$ under mild assumption:

**Theorem 2.** *Suppose that $\boldsymbol{\lambda}$ is updated with increasing simulation step $t$, and the search tree is reset at the end of $\boldsymbol{\lambda}$'s update as detailed in Appendix F. If the asymptotic bias of UCT holds for all types of cost values (i.e. $\exists M > 0, \forall k, |V_{C_k}^{\pi_{\boldsymbol{\lambda}}^*}(h_0) - V_{C_k}(h_0)| \leq M(\frac{\log t}{t}))$, then either $sign(V_{C_k}^{\pi_{\boldsymbol{\lambda}}^*}(h_0) - \hat{c}_k) = sign(V_{C_k}(h_0) - \hat{c}_k)$ or $|V_{C_k}^{\pi_{\boldsymbol{\lambda}}^*}(h_0) - \hat{c}_k| \leq M\frac{\log t}{t}$ holds with probability 1 as $t \to \infty$.*

The above states that either $\boldsymbol{\lambda}$ is close to optimal or it is improved by the update towards the direction of negative subgradient. Note that CC-POMCP inherits the scalability of POMCP and thus does not require an explicit model of the environment: all we need is a black-box simulator $\mathcal{G}$ of the CPOMDP, which generates sample $(s', o, r, c) \sim \mathcal{G}(s, a)$ of the next state $s'$, observation $o$, reward $r$, and cost vector $c$, given the current state $s$ and action $a$.

### 4.1 Admissible Costs

After the agent executes an action, the cost constraint threshold $\hat{c}$ must be updated at the next time step. For this, we reformulate the notion of *admissible cost* [17], originally formulated for dynamic programming. The admissible cost $\hat{\mathbf{c}}_{t+1}$ at time step $t + 1$ denotes the expected total cost allowed to be incurred in future time steps $\{t + 1, t + 2, ...\}$ without violating the cost constraints. Under dynamic programming, the update is given by [17]: $\hat{\mathbf{c}}_{t+1} = \frac{\hat{\mathbf{c}}_t - \mathbb{E}[\mathbf{C}(b_t, a_t)|b_0, \pi]}{\gamma}$ where evaluating $\mathbb{E}[\mathbf{C}(b_t, a_t)|b_0, \pi]$ requires the probability of reaching $(b_t, a_t)$ at time step $t$, which in turn requires marginalizing out the history in the past $[a_0, b_1, a_2, ..., b_{t-1}]$. This is intractable for large state spaces.

On the other hand, under forward search, the admissible cost at the next time step $t + 1$ is simply $\hat{\mathbf{c}}_{t+1} = V_{\mathbf{C}}^{\pi^*}(b_{t+1})$. We can access $V_{\mathbf{C}}^{\pi^*}(b_{t+1})$ by starting from the root node of the search tree $h_t$, and sequentially following the action branch $a_t$ and the next observation branch $o_{t+1}$. Here we are assuming that the exact optimal $V_{\mathbf{C}}^{\pi^*}$ is obtained, which is certainly achievable after infinitely many simulations of CC-POMCP. Note also that even though $\hat{\mathbf{c}}_t > V_{\mathbf{C}}^{\pi^*}(b_t)$ is possible in general, assuming $\hat{\mathbf{c}}_t = V_{\mathbf{C}}^{\pi^*}(b_t)$ does not change the solution. If $\hat{\mathbf{c}}_t < V_{\mathbf{C}}^{\pi^*}(b_t)$, this means that no feasible policy exists.

### 4.2 Filling the Gap: Stochastic vs Deterministic Policies

Our approach relies on the POMDP with scalarized rewards, but care must be taken as the optimal policy of the CPOMDP is generally stochastic: given optimal $\boldsymbol{\lambda}^*$, let $\pi_{\boldsymbol{\lambda}}^*$ be the *deterministic* optimal policy for the POMDP with the scalarized reward function $R - \boldsymbol{\lambda}^{*\top}\mathbf{C}$. Then, by the duality between the primal (1) and the dual (2),

$$V_R^*(b_0; \hat{\mathbf{c}}) = \mathcal{V}_{\boldsymbol{\lambda}}^*(b_0) + \boldsymbol{\lambda}^{*\top}\hat{\mathbf{c}} = V_R^{\pi_{\boldsymbol{\lambda}}^*}(b_0) - \boldsymbol{\lambda}^{*\top}(V_{\mathbf{C}}^{\pi_{\boldsymbol{\lambda}}^*}(b_0) - \hat{\mathbf{c}}) \tag{8}$$

This implies that if $\lambda_k^* > 0$ and $V_{C_k}^{\pi_{\boldsymbol{\lambda}}^*}(b_0) \neq c_k$ for some $k$ then $\pi_{\boldsymbol{\lambda}}^*$ is not optimal for the original CPOMDP. This is exactly the situation where the optimal policy is stochastic. In order to make the policy computed by our algorithm stochastic, we make sure that the following *optimality condition* is satisfied, derived from $V_R^*(b_0; \hat{\mathbf{c}}) = V_R^{\pi_{\boldsymbol{\lambda}}^*}(b_0)$:

$$\sum_{k=1}^{K} \lambda_k^*(V_{C_k}^{\pi_{\boldsymbol{\lambda}}^*}(b_0) - \hat{c}_k) = \sum_{k=1}^{K} \lambda_k^* \left( \sum_a \pi_{\boldsymbol{\lambda}}^*(a|b_0)Q_{C_k}^{\pi_{\boldsymbol{\lambda}}^*}(b_0, a) - \hat{c}_k \right) = 0 \tag{9}$$

That is, actions $a_i^*$ with equally maximal scalarized action values $\mathcal{Q}_{\boldsymbol{\lambda}}(b, a_i^*) = Q_R(b, a_i^*) - \boldsymbol{\lambda}^{\top}Q_{\mathbf{C}}(b, a_i^*)$ participate as the support of the stochastic policy, and are selected with probability $\pi(a_i^*|b)$ that satisfies $\forall k : \lambda_k^* > 0, \sum_{a_i^*} \pi(a_i^*|b)Q_{C_k}(b, a_i^*) = \hat{c}_k$.

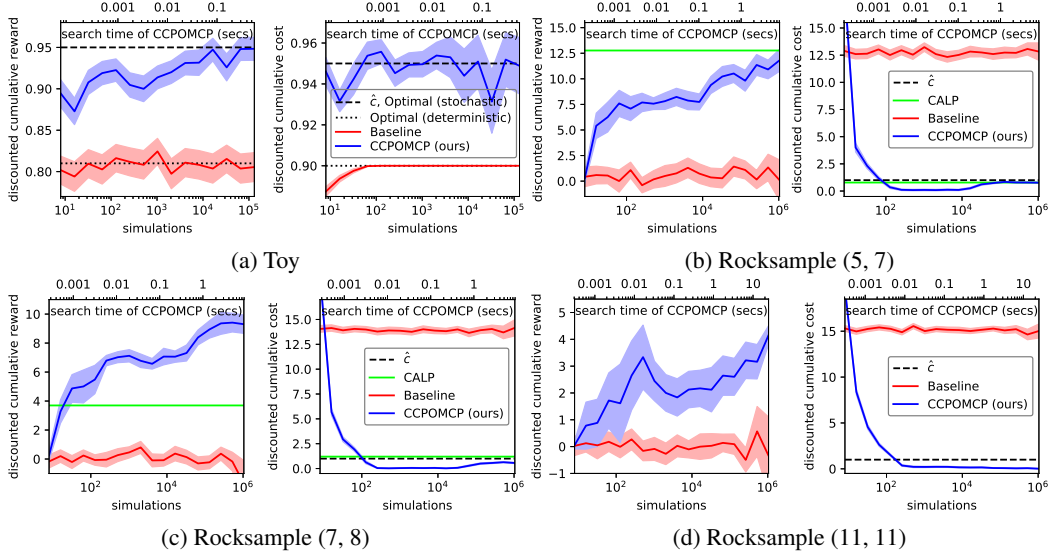

Figure 1: The result of CPOMDP Toy domain [11] and the constrained variants of *Rocksample* [21]. The result of *Rocksample* (15, 15) is presented in Appendix I. For each domain, the left figure shows the average discounted cumulative reward, and the right figure shows the average discounted cumulative cost. The wall-clock search time for CC-POMCP is presented on the top of $x$-axis.

GREEDYPOLICY in Algorithm 1 computes the stochastic policy according to the above principle. In practice, due to the randomness in Monte-Carlo sampling, action values in (9) are always subject to estimation error, so it is reformulated as a linear programming with up to $|A| + 2K$ variables:

$$\min_{\{w_i, \xi_k^+, \xi_k^-\}} \sum_{k=1}^{K} \lambda_k (\xi_k^+ + \xi_k^-) \tag{10}$$

$$s.t. \sum_{i:a_i^* \in A^*} w_i Q_{C_k}(h, a_i^*) = \hat{c}_k + (\xi_k^+ - \xi_k^-) \quad \forall k$$

$$\sum_{i:a_i^* \in A^*} w_i = 1 \ \text{ and } \ w_i, \xi_k^+, \xi_k^- \geq 0$$

where $A^* = \{a_i^* \mid \mathcal{Q}_{\boldsymbol{\lambda}}(h, a_i^*) \simeq \mathcal{Q}_{\boldsymbol{\lambda}}(h, a^*) \text{ s.t. } a^* = \arg\max_a \mathcal{Q}_{\boldsymbol{\lambda}}^{\oplus}(h, a)\}$ [2], and the solutions are $w_i = \pi(a_i^*|h)$. Here, when $K = 1$, there is a simple analytic solution to LP (10), which is described in Appendix G. Even when $K > 1$, note that the optimization problem occurs only when the number of equally maximal scalarized action values is more than 1 thus randomization of actions is required. It is well known in CMDPs that an optimal policy requires at most $K$ randomizations [1], which means that we expect to invoke optimization on extremely small part of the state space when the problem is very large.

## 5 Experiments

All the parameters for running CC-POMCP are provided in Appendix H.

**Baseline agent** To the best of our knowledge, this work is the first attempt to solve *constrained* (PO)MDP using Monte-Carlo Tree Search. Since there is no algorithm for direct performance comparison for large problems, we implemented a simple baseline agent using MCTS. This agent works as outlined in section 2: it chooses an action via reward-maximizing POMCP while preventing action branches that violate cost constraint $Q_C(s, a) \leq \hat{c}$. If all action branches violate the cost constraints, the agent chooses action uniformly at random.

| Domain | $|S|$ | $\hat{c}$ | Algorithm | Cumulative reward | Cumulative cost |
|---|---|---|---|---|---|
| Rocksample (5,7) | 3,201 | 1 | CALP | **12.77±0** | **0.78±0** |
| | | | Baseline | 1.09±0.88 | 12.74±0.50 |
| | | | CC-POMCP | 11.36±1.02 | 0.79±0.06 |
| Rocksample (7,8) | 12,544 | 1 | CALP | 3.67±0 | 1.20±0 |
| | | | Baseline | −0.23±0.44 | 13.92±0.33 |
| | | | CC-POMCP | **9.36±0.76** | **0.56±0.06** |
| Rocksample (11,11) | 247,808 | 1 | CALP | N/A | N/A |
| | | | Baseline | 0.14±0.33 | 15.29±0.25 |
| | | | CC-POMCP | **2.65±0.73** | **0.09±0.04** |
| Rocksample (15,15) | 7,372,800 | 1 | CALP | N/A | N/A |
| | | | Baseline | 0.39±0.58 | 16.27±0.27 |
| | | | CC-POMCP | **0.74±0.33** | **0.69±0.08** |

Table 1: Comparison of CC-POMDP with the state-of-the-art offline solver, CALP [18].

**CPOMDP: Toy and Rocksample**   We first tested CC-POMCP on the synthetic toy domain introduced in [11] to demonstrate convergence to stochastic optimal actions, where the cost constraint $\hat{c}$ is 0.95. Any deterministic policy is suboptimal or violates the cost constraint. As can be seen in Figure 1a, CC-POMCP converges to optimal stochastic action selection (thus experimentally confirms the soundness of algorithm), while the baseline agent converges to the suboptimal policy (optimal policy among deterministic ones).

We also conducted experiments on cost-constrained variants of *Rocksample* [21]. *Rocksample*$(n, k)$ simulates a Mars rover in $n \times n$ grid containing $k$ rocks. The goal is to sort out good rocks, collect them, and escape the map by moving to the rightmost part of the map. We augmented the single-objective *Rocksample* with the cost function that assigns 1 to low reward state-action pairs (i.e. $C_p(s, a) = 1$ if $R_p(s, a) < 0$), similarly to [18]. We also assigned the cost of 1 to actions detecting whether a rock is good or bad. The cost constraint $\hat{c}$ is set to 1. We compared CC-POMCP with the state-of-the-art offline CPOMDP solver, CALP [18]. CALP was allowed 10 minutes for the offline computation, and we performed exact policy evaluation with respect to the resulting finite state controller without simulation in the real environment. The results on *Rocksample* are summarized in Table 1 and Figure 1. In *Rocksample* (5, 7), the reward performance of CC-POMCP is comparable to CALP when more than 2 seconds of search time is allowed while at the same time satisfying the cost constraint. In contrast, baseline agent basically exhibited random behavior since the Monte-Carlo return at early stage mostly violates cost constraints for all actions. On *Rocksample (7, 8)*, CALP failed to compute a feasible policy, and CC-POMCP outperformed CALP in terms of reward while satisfying the cost constraint. Finally, CC-POMCP was able to scale to *Rocksample (11, 11)* and *(15, 15)*: given a few seconds of search time, CC-POMCP was able to find actions satisfying the cost constraints, and tended to yield higher returns as we increased the number of simulations.

**CMDP: Pong**   We also conducted experiments on a multi-objective version of PONG, an arcade game running on the Arcade Learning Environment (ALE) [3], depicted in Figure 2a. In this domain, the left paddle is handled by the default computer opponent and the right paddle is controlled by the agent. We use the RAM state feature, i.e. the states are binary strings of length 1024 which results in $|S| = 2^{1024}$. The action space is {up, down, stay}. The agent receives a reward of $\{1, -1\}$ for each round depending on win/lose. The episode terminates if the accumulated reward is $\{21, -21\}$. We assigned cost 0 to the center area ($position \in [0.4, 0.6]$), 1 to the neighboring area ($position \in [0.2, 0.4] \cup [0.6, 0.8]$), and 2 to the area farthest away from the center ($position \in [0.0, 0.2] \cup [0.8, 1.0]$). This cost function was motivated by the scenario, where a human expert tries to constrain the RL agent to adhere to human advice that the agent should stay in the center. This advice is encoded as the cost function and its threshold. We experimented with various cost constraint thresholds $\hat{c} \in \{200, 100, 50, 30, 20\}$ ranging from the unconstrained case $\hat{c} = 200$ ($\because \frac{C_{\max}}{1-\gamma} = 200$) to the tightly constrained case $\hat{c} = 20$. We can see that the agent has two conflicting objectives: in order to achieve high rewards, it sometimes needs to move the paddle to positions far away from the center, but if this happens too often, the cost constraint will be violated. Thus, it needs to trade off between reward and cost properly depending on the cost constraint threshold $\hat{c}$.

Figure 2b summarizes the experimental results from the CC-POMCP and the baseline agents. When $\hat{c} = 200$ (unconstrained case), both algorithms always win the game 21 by 0. As we lower $\hat{c}$,

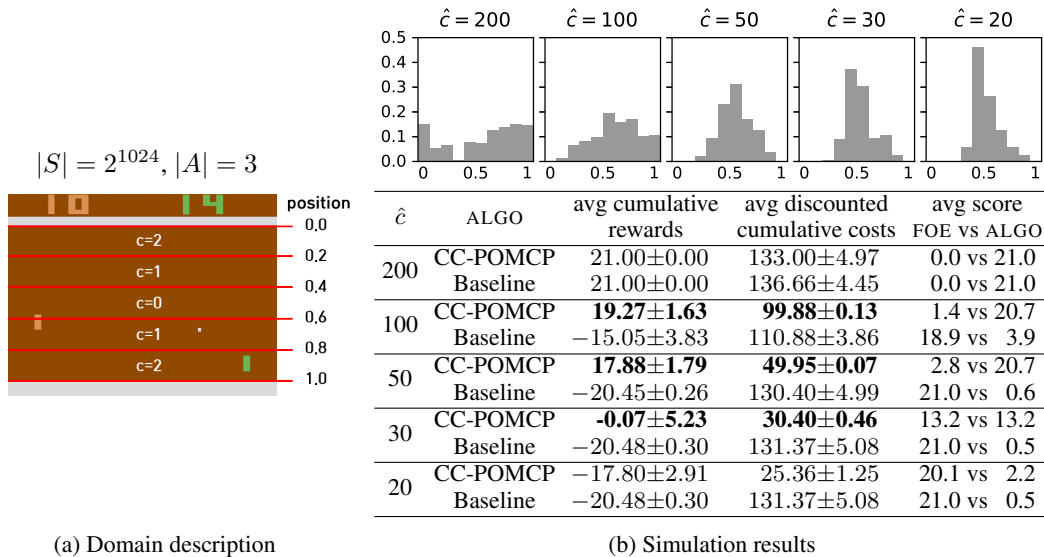

$|S| = 2^{1024}, |A| = 3$

| $\hat{c}$ | ALGO | avg cumulative rewards | avg discounted cumulative costs | avg score FOE vs ALGO |
|---|---|---|---|---|
| 200 | CC-POMCP | 21.00±0.00 | 133.00±4.97 | 0.0 vs 21.0 |
| | Baseline | 21.00±0.00 | 136.66±4.45 | 0.0 vs 21.0 |
| 100 | CC-POMCP | **19.27±1.63** | **99.88±0.13** | 1.4 vs 20.7 |
| | Baseline | −15.05±3.83 | 110.88±3.86 | 18.9 vs 3.9 |
| 50 | CC-POMCP | **17.88±1.79** | **49.95±0.07** | 2.8 vs 20.7 |
| | Baseline | −20.45±0.26 | 130.40±4.99 | 21.0 vs 0.6 |
| 30 | CC-POMCP | **-0.07±5.23** | **30.40±0.46** | 13.2 vs 13.2 |
| | Baseline | −20.48±0.30 | 131.37±5.08 | 21.0 vs 0.5 |
| 20 | CC-POMCP | −17.80±2.91 | 25.36±1.25 | 20.1 vs 2.2 |
| | Baseline | −20.48±0.30 | 131.37±5.08 | 21.0 vs 0.5 |

(a) Domain description              (b) Simulation results

Figure 2: (a) Multi-objective version of Atari 2600 PONG, visualizing the cost function. (b) Results of the constrained PONG. Above: Histogram of the CC-POMCP agent's position, where the horizontal axis denotes the position of the agent (0: topmost, 1: bottommost) and the vertical axis denotes the relative discounted visitation rate for each bin.

CC-POMCP tends to stay in the center in order to make a trade off between reward and cost (shown in the histograms in Figure 2b). We can also see that the agent gradually performs worse in terms of scores as $\hat{c}$ decreases. This is a natural result since it is forced to stay in the center and thus sacrifice the game score. Overall, CC-POMCP computes a good policy while generally respecting the cost constraint. On the other hand, the baseline fails to show a meaningful policy except when $\hat{c} = 200$ since the Monte-Carlo cost returns at early stage mostly violate the cost constraint, resulting in random behavior.

# 6 Conclusion

We presented CC-POMCP, an online MCTS algorithm for very large CPOMDPs. We showed that solving the dual LP of CPOMDPs is equivalent to jointly solving an unconstrained POMDP and optimizing its LP-induced parameters $\boldsymbol{\lambda}$, and provided theoretical results that shed insight on the properties of $\boldsymbol{\lambda}$ and how to optimize it. We then extended POMCP to maximize the scalarized value while simultaneously updating $\boldsymbol{\lambda}$ using the current action-value estimates $Q_{\mathbf{C}}$. We also empirically showed that CC-POMCP converges to the optimal stochastic actions on a toy domain and easily scales to very large CPOMDPs through the constrained variants of *Rocksample* and the multi-objective version of PONG.

## Acknowledgement

This work was supported by the ICT R&D program of MSIT/IITP of Korea (No. 2017-0-01778) and DAPA/ADD of Korea (UD170018CD). J. Lee acknowledges the Global Ph.D. Fellowship Program by NRF of Korea (NRF-2018-Global Ph.D. Fellowship Program).

## Footnotes

[1] Stochastic nature of the optimal policy in CPOMDPs results from the stochasticity of optimal policies in CMDPs [1]. A more formal treatment on this matter, pertinent to CPOMDPs, can be found in [6].

[2]Exact condition for $\mathcal{Q}_{\boldsymbol{\lambda}}(h, a_i^*) \simeq \mathcal{Q}_{\boldsymbol{\lambda}}(h, a^*)$ and its theoretical guarantee are provided in Appendix E.

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
