[Supplementary Material]

## Supplementary Material: Monte-Carlo Tree Search for Constrained POMDPs (Jongmin Lee, Geon-Hyeong Kim, Pascal Poupart, Kee-Eung Kim)

## Appendix A  Proof of Proposition 1

**Proposition 1.** *Let $\mathcal{V}_{\boldsymbol{\lambda}}^*$ be the optimal value function of the POMDP with scalarized reward function $R(b,a) - \boldsymbol{\lambda}^\top \mathbf{C}(b,a)$. Then, $\mathcal{V}_{\boldsymbol{\lambda}}^*(b_0) + \boldsymbol{\lambda}^\top \hat{\mathbf{c}}$ is a piecewise-linear and convex (PWLC) function over $\boldsymbol{\lambda}$.*

*Proof.* We give a proof by induction. For all $b$,

$$\mathcal{V}_{\boldsymbol{\lambda}}^{(0)}(b) = \max_a \left[ R(b,a) - \boldsymbol{\lambda}^\top \mathbf{C}(b,a) \right]$$

is a piecewise-linear and convex function over $\boldsymbol{\lambda}$ since the max of linear functions is piecewise linear and convex. Now, assume the following induction hypothesis: $\mathcal{V}_{\boldsymbol{\lambda}}^{(k)}(b)$ is piecewise-linear and convex function over $\boldsymbol{\lambda}$ for all $b$. Then, for all $b$,

$$\mathcal{V}_{\boldsymbol{\lambda}}^{(k+1)}(b) = \max_a \Big[ \underbrace{R(b,a) - \boldsymbol{\lambda}^\top \mathbf{C}(b,a)}_{\text{linear in } \boldsymbol{\lambda}} + \gamma \sum_{o,s,s'} \underbrace{Z_p(o|s',a)T_p(s'|s,a)b(s)\mathcal{V}_{\boldsymbol{\lambda}}^{(k)}(b^{ao})}_{\text{PWLC in } \boldsymbol{\lambda}} \Big]$$

is also PWLC since the summation of PWLC functions is PWLC and max over PWLC functions is again PWLC. As a consequence, $\mathcal{V}_{\boldsymbol{\lambda}}^*(b_0) = \lim_{k\to\infty} \mathcal{V}_{\boldsymbol{\lambda}}^{(k)}(b_0)$ is PWLC over $\boldsymbol{\lambda}$ and so is $\mathcal{V}_{\boldsymbol{\lambda}}^*(b_0) + \boldsymbol{\lambda}^\top \hat{\mathbf{c}}$. □

Figure 3: $\mathcal{V}_{\boldsymbol{\lambda}}^*(b_0) + \boldsymbol{\lambda}^\top \hat{\mathbf{c}}$ for a simple CPOMDP, which is piecewise-linear and convex. Red line represents the trajectory of $\boldsymbol{\lambda}$ starting from $\boldsymbol{\lambda} = [0,0]^\top$ and sequentially updated by $\boldsymbol{\lambda} \leftarrow \boldsymbol{\lambda} + \alpha_n(V_{\mathbf{C}}^{\pi_{\boldsymbol{\lambda}}^*}(b_0) - \hat{\mathbf{c}})$.

## Appendix B  Proof of Lemma 1

**Lemma 1.** *Let $\mathcal{M}_1 = \langle B, A, T, R_1, \gamma \rangle$ and $\mathcal{M}_2 = \langle B, A, T, R_2, \gamma \rangle$ be two (belief-state) MDPs differing only in the reward function, and $V_1^\pi$ and $V_2^\pi$ be the value functions of $\mathcal{M}_1$ and $\mathcal{M}_2$ with a fixed policy $\pi$. Then, the value function of the new MDP $\mathcal{M} = \langle B, A, T, pR_1 + qR_2, \gamma \rangle$ with the policy $\pi$ is $V^\pi(b) = pV_1^\pi(b) + qV_2^\pi(b)$ for all $b \in B$.*

*Proof.* We give a proof by induction. For all $b$,

$$V^{(0)}(b) = \sum_a \pi(a|b) \left[ pR_1(b,a) + qR_2(b,a) \right]$$

$$= p \sum_a \pi(a|b)R_1(b,a) + q \sum_a \pi(a|b)R_2(b,a)$$

$$= pV_1^{(0)}(b) + qV_2^{(0)}(b)$$

Then, assume the induction hypothesis $V^{(k)}(b) = pV_1^{(k)}(b) + qV_2^{(k)}(b)$. For all $b$,

$$V^{(k+1)}(b)$$

$$= \sum_a \pi(a|b)\left[pR_1(b,a) + qR_2(b,a) + \gamma\sum_{b'}T(b'|b,a)V^{(k)}(b')\right]$$

$$= p\sum_a \pi(a|b)\left[R_1(b,a) + \gamma\sum_{b'}T(b'|b,a)V_1^{(k)}(b')\right]$$

$$+ q\sum_a \pi(a|b)\left[R_2(b,a) + \gamma\sum_{b'}T(b'|b,a)V_2^{(k)}(b')\right]$$

$$= pV_1^{(k+1)}(b) + qV_2^{(k+1)}(b)$$

As a consequence, $V^\pi(b) = \lim_{k\to\infty}V^{(k)}(b) = \lim_{k\to\infty}(V_1^{(k)}(b) + V_2^{(k)}(b)) = V_1^\pi(b) + V_2^\pi(b)$. $\square$

## Appendix C  Proof of Theorem 1

**Theorem 1.** *For any $\boldsymbol{\lambda}$, $V_{\mathbf{C}}^{\pi_{\boldsymbol{\lambda}}^*}(b_0) - \hat{\mathbf{c}}$ is a negative subgradient that decreases the objective in Eq. (3), where $\pi_{\boldsymbol{\lambda}}^*$ is the optimal policy with respect to the scalarized reward function $R(b,a) - \boldsymbol{\lambda}^\top \mathbf{C}(b,a)$. Also, if $V_{\mathbf{C}}^{\pi_{\boldsymbol{\lambda}}^*}(b_0) - \hat{\mathbf{c}} = \mathbf{0}$ then $\boldsymbol{\lambda}$ is the optimal solution of Eq. (3).*

*Proof.* For any $\boldsymbol{\lambda}_0$ and $\boldsymbol{\lambda}_1$,

$$(\mathcal{V}_{\boldsymbol{\lambda}_1}^*(b_0) + \boldsymbol{\lambda}_1^\top\hat{\mathbf{c}}) - (\mathcal{V}_{\boldsymbol{\lambda}_0}^*(b_0) + \boldsymbol{\lambda}_0^\top\hat{\mathbf{c}})$$
$$=(V_R^{\pi_{\boldsymbol{\lambda}_1}^*}(b_0) - \boldsymbol{\lambda}_1^\top V_{\mathbf{C}}^{\pi_{\boldsymbol{\lambda}_1}^*}(b_0)) - (V_R^{\pi_{\boldsymbol{\lambda}_0}^*}(b_0) - \boldsymbol{\lambda}_0^\top V_{\mathbf{C}}^{\pi_{\boldsymbol{\lambda}_0}^*}(b_0)) + (\boldsymbol{\lambda}_1 - \boldsymbol{\lambda}_0)^\top\hat{\mathbf{c}} \qquad \text{(Lemma 1)}$$
$$\geq(V_R^{\pi_{\boldsymbol{\lambda}_0}^*}(b_0) - \boldsymbol{\lambda}_1^\top V_{\mathbf{C}}^{\pi_{\boldsymbol{\lambda}_0}^*}(b_0)) - (V_R^{\pi_{\boldsymbol{\lambda}_0}^*}(b_0) - \boldsymbol{\lambda}_0^\top V_{\mathbf{C}}^{\pi_{\boldsymbol{\lambda}_0}^*}(b_0)) + (\boldsymbol{\lambda}_1 - \boldsymbol{\lambda}_0)^\top\hat{\mathbf{c}}$$
$$(\because \pi_{\boldsymbol{\lambda}_1}^* \text{ is optimal w.r.t. } R - \boldsymbol{\lambda}_1^\top\mathbf{C})$$
$$=(\boldsymbol{\lambda}_1 - \boldsymbol{\lambda}_0)^\top(\hat{\mathbf{c}} - V_{\mathbf{C}}^{\pi_{\boldsymbol{\lambda}_0}^*}(b_0)) \qquad\qquad\qquad (11)$$

Therefore, $\hat{\mathbf{c}} - V_{\mathbf{C}}^{\pi_{\boldsymbol{\lambda}_0}^*}(b_0)$ is a *subgradient* of $\mathcal{V}_{\boldsymbol{\lambda}}^*(b_0) + \boldsymbol{\lambda}^\top\hat{\mathbf{c}}$ at point $\boldsymbol{\lambda} = \boldsymbol{\lambda}_0$, which concludes that $V_{\mathbf{C}}^{\pi_{\boldsymbol{\lambda}_0}^*}(b_0) - \hat{\mathbf{c}}$ is a *negative subgradient* at point $\boldsymbol{\lambda} = \boldsymbol{\lambda}_0$.

In addition, suppose $\boldsymbol{\lambda}_1 = \boldsymbol{\lambda}_0 + \alpha(V_{\mathbf{C}}^{\pi_{\boldsymbol{\lambda}_0}^*}(b_0) - \hat{\mathbf{c}})$. Then, for sufficiently small $\alpha > 0$, the new reward function $R - \boldsymbol{\lambda}_1^\top\mathbf{C}$ which is slightly changed from the old reward $R - \boldsymbol{\lambda}_0^\top\mathbf{C}$ still satisfies the *reward optimality condition* with respect to policy $\pi_{\boldsymbol{\lambda}_0}^*$ [15]. In this situation, $\pi_{\boldsymbol{\lambda}_0}^* = \pi_{\boldsymbol{\lambda}_1}^*$ and we obtain:

$$(\mathcal{V}_{\boldsymbol{\lambda}_0}^*(b_0) + \boldsymbol{\lambda}_0^\top\hat{\mathbf{c}}) - (\mathcal{V}_{\boldsymbol{\lambda}_1}^*(b_0) + \boldsymbol{\lambda}_1^\top\hat{\mathbf{c}}) \geq (\boldsymbol{\lambda}_0 - \boldsymbol{\lambda}_1)^\top(\hat{\mathbf{c}} - V_{\mathbf{C}}^{\pi_{\boldsymbol{\lambda}_1}^*}(b_0)) \qquad \text{(by result of (11))}$$
$$= -\alpha(V_{\mathbf{C}}^{\pi_{\boldsymbol{\lambda}_0}^*}(b_0) - \hat{\mathbf{c}})^\top(\hat{\mathbf{c}} - V_{\mathbf{C}}^{\pi_{\boldsymbol{\lambda}_1}^*}(b_0))$$
$$= \alpha\|\hat{\mathbf{c}} - V_{\mathbf{C}}^{\pi_{\boldsymbol{\lambda}_1}^*}(b_0))\|_2^2$$
$$\geq 0$$

Also, if $V_{\mathbf{C}}^{\pi_{\boldsymbol{\lambda}}^*}(b_0) - \hat{\mathbf{c}} = \mathbf{0}$, then the dual objective in Eq. (4) becomes $V_R^{\pi_{\boldsymbol{\lambda}}^*}(b_0)$. By the weak duality theorem, $V_R^{\pi_{\boldsymbol{\lambda}}^*}(b_0) \geq \sum_{b,a}R(b,a)y^*(b,a) = V_R^*(b_0;\hat{\mathbf{c}})$ holds where $y^*(b,a)$ is the optimal solution of the primal LP (1). Assuming that $V_{\mathbf{C}}^{\pi_{\boldsymbol{\lambda}}^*}(b_0) = \hat{\mathbf{c}}$, $\pi_{\boldsymbol{\lambda}}^*$ is a feasible policy that satisfies the cost constraints, and $V_R^{\pi_{\boldsymbol{\lambda}}^*}(b_0)$ cannot be larger than $V_R^*(b_0;\hat{\mathbf{c}})$. That is, $V_R^{\pi_{\boldsymbol{\lambda}}^*}(b_0) = V_R^*(b_0;\hat{\mathbf{c}})$ and the duality gap is zero, which means that $\boldsymbol{\lambda}$ is the optimal solution. $\square$

## Appendix D  Recursive Update of Admissible Cost

As an alternative to using $V_C^*(h_{t+1})$, we can also do recursive update of the admissible cost at time step $t+1$ to depend only on the current expected immediate cost at $(h_t, a_t)$ and the current optimal cost value $Q_{\mathbf{C}}^{\pi^*}(h_t, a_t)$ by the following relationship.

$$\hat{\mathbf{c}}_t = V_{\mathbf{C}}^{\pi^*}(h_t) = \sum_a \pi^*(a|h_t) Q_{\mathbf{C}}^{\pi^*}(h_t, a)$$

$$= \pi^*(a_t|h_t) Q_{\mathbf{C}}^{\pi^*}(h_t, a_t) + \sum_{a \neq a_t} \pi^*(a|h_t) Q_{\mathbf{C}}^{\pi^*}(h_t, a)$$

$$= \pi^*(a_t|h_t) \left[ \mathbf{C}(h_t, a_t) + \gamma \underbrace{\mathbb{E}[V_{\mathbf{C}}^{\pi^*}(h_{t+1})|h_t, a_t]}_{=\hat{\mathbf{c}}_{t+1}} \right] + \sum_{a \neq a_t} \pi^*(a|h_t) Q_{\mathbf{C}}^{\pi^*}(h_t, a)$$

$$\therefore \hat{\mathbf{c}}_{t+1} = \frac{\hat{\mathbf{c}}_t - \pi^*(a_t|h_t)\mathbf{C}(h_t, a_t) - \sum_{a \neq a_t} \pi^*(a|h_t) Q_{\mathbf{C}}^{\pi^*}(h_t, a)}{\gamma \pi(a_t|b_t)} \qquad (12)$$

We used Eq. (12) to update the admissible cost in the experiments.

## Appendix E  Equality Test for Collecting Optimal Action Candidates

---
**Algorithm 2** SEARCH of CC-POMCP

---
1: **function** SEARCH($h_0$)
2:     $\lambda$ is randomly initialized
3:     **for** $n = 1, 2, \ldots$ **do**
4:         **for** $t = 1, 2, \ldots, f(n)$ **do**     # $f(n)$ is any monotonically increasing sequence w.r.t. $n$.
5:             **if** $h = \emptyset$ **then**
6:                 $s \sim b_0$
7:             **else**
8:                 $s \sim \mathcal{B}(h_0)$
9:             **end if**
10:             SIMULATE($s, h_0, 0$)
11:         **end for**
12:         $a \sim$ GREEDYPOLICY($h_0, 0, 0$)
13:         $\lambda \leftarrow \lambda + \alpha_n [Q_C(h_0, a) - \hat{c}]$
14:         Clip $\lambda_k$ to range $[0, \frac{R_{\max} - R_{\min}}{\tau(1-\gamma)}]$   $\forall k = \{1, 2, \ldots K\}$
15:         Reset entire search tree $T(h_0)$
16:     **end for**
17:     **return** GREEDYPOLICY($h_0, 0, \nu$)
18: **end function**

---

To compute a stochastic optimal policy, we first need to collect the candidates of the support of the optimal actions. Since there always exists some error for estimating $\mathcal{Q}_\lambda(h, a)$ due to the randomness of Monte-Carlo sampling, we need an explicit criterion to determine whether an action $\hat{a}$ can be treated as an optimum (i.e. $\mathcal{Q}_\lambda(h, \hat{a}) \simeq Q_\lambda(h, a^*)$). The following theorem provides the *equality test* criterion to collect optimal action candidates and its validity.

**Theorem 3.** *Let $\mathcal{Q}_\lambda$ be the scalarized action-value estimated by CC-POMCP using SEARCH in Algorithm 2, and $a^* = \arg\max_a \mathcal{Q}_\lambda^\oplus(h, a)$. For any $\nu > 0$, suppose we use the following equality test criterion for the optimal action selection: $|\mathcal{Q}_\lambda(h, a^*) - \mathcal{Q}_\lambda(h, \hat{a})| \leq \nu \left[ \sqrt{\frac{\log N(h, a^*)}{N(h, a^*)}} + \sqrt{\frac{\log N(h, \hat{a})}{N(h, \hat{a})}} \right]$. Then it accepts all of the optimal actions (with respect to MDP with the scalarized reward function $R(s, a) - \lambda^\top \mathbf{C}(s, a)$) while rejecting all of the suboptimal actions with probability 1 as $t \to \infty$.*

To prove Theorem 3, we first introduce a lemma from [13]:

**Lemma 2** (Theorem 7 in [13]). *When the UCT algorithm is running on a tree with depth $D$, the bias of expected payoff is $O((|A|D \log t + |A|^D)/t)$ after $t$ iteration. Moreover, the probability that UCT algorithm fails to select optimal action at root converges to zero as $t \to \infty$.*

By Lemma 2, there exists $M > 0$ such that the following holds:

$$\Pr\left(|\mathcal{Q}^*_{\boldsymbol{\lambda}}(h,a) - \mathcal{Q}_{\boldsymbol{\lambda}}(h,a)| \geq M \frac{\log N(h,a)}{N(h,a)}\right) \to 0 \text{ as } N(h,a) \to \infty, \tag{13}$$

where $\mathcal{Q}^*_{\boldsymbol{\lambda}} = \mathcal{Q}^{\pi^*_{\boldsymbol{\lambda}}}_{\boldsymbol{\lambda}}$ (true optimal value function of MDP with the scalarized reward function). In order to prove Theorem 3, we first provide the following two lemmas based on Eq. (13).

**Lemma 3.** *Let $a^* = \arg\max_a \mathcal{Q}^\oplus_{\boldsymbol{\lambda}}(h,a)$, and $\hat{a}$ be a suboptimal action (with respect to MDP with the scalarized reward function $R(s,a) - \boldsymbol{\lambda}^\top \mathbf{C}(s,a)$). For the given $\boldsymbol{\lambda}$, if $a^*$ is an optimal action, then $\Pr\left(|\mathcal{Q}_{\boldsymbol{\lambda}}(h,a^*) - \mathcal{Q}_{\boldsymbol{\lambda}}(h,\hat{a})| \leq \nu\left[\sqrt{\frac{\log N(h,a^*)}{N(h,a^*)}} + \sqrt{\frac{\log N(h,\hat{a})}{N(h,\hat{a})}}\right]\right) \to 0$ as $t \to \infty$. In other words, all of the suboptimal actions are rejected asymptotically by the proposed equality test criterion.*

*Proof.* Let $\mathcal{Q}^*_{\boldsymbol{\lambda}}(h,a^*) - \mathcal{Q}^*_{\boldsymbol{\lambda}}(h,\hat{a}) = \Delta > 0$, where $\mathcal{Q}^*_{\boldsymbol{\lambda}} = \mathcal{Q}^{\pi^*_{\boldsymbol{\lambda}}}_{\boldsymbol{\lambda}}$. Then using the fact that $\Pr(A+B < C+D) \leq \Pr(A < C) + \Pr(B < D)$, we can derive the following inequality:

$$\Pr\left(\mathcal{Q}_{\boldsymbol{\lambda}}(h,a^*) - \mathcal{Q}_{\boldsymbol{\lambda}}(h,\hat{a}) \leq \nu\left[\sqrt{\frac{\log N(h,a^*)}{N(h,a^*)}} + \sqrt{\frac{\log N(h,\hat{a})}{N(h,\hat{a})}}\right]\right)$$

$$\leq \Pr\left(\mathcal{Q}_{\boldsymbol{\lambda}}(h,a^*) - \mathcal{Q}^*_{\boldsymbol{\lambda}}(h,a^*) + \frac{\Delta}{2} \leq \nu\sqrt{\frac{\log N(h,a^*)}{N(h,a^*)}}\right)$$

$$+ \Pr\left(\mathcal{Q}^*_{\boldsymbol{\lambda}}(h,\hat{a}) - \mathcal{Q}_{\boldsymbol{\lambda}}(h,\hat{a}) + \frac{\Delta}{2} \leq \nu\sqrt{\frac{\log N(h,\hat{a})}{N(h,\hat{a})}}\right). \tag{14}$$

Therefore, it is enough to show that both of two terms in the right-hand side of (14) converge to 0 as $t \to \infty$. It is obvious that the first term converges to probability 0 since $\mathcal{Q}_{\boldsymbol{\lambda}}(h,a^*) - \mathcal{Q}^*_{\boldsymbol{\lambda}}(h,a^*) \to 0$ by Eq. (13) and $\sqrt{\frac{\log N(h,a^*)}{N(h,a^*)}} \to 0$ as $N(h,a^*) \to \infty$. The second term also converges to probability 0 by the same reasoning. $\square$

**Lemma 4.** *Let $a^* = \arg\max_a \mathcal{Q}^\oplus_{\boldsymbol{\lambda}}(h,a)$, and $\hat{a}$ be another optimal action (with respect to MDP with the scalarized reward function $R(s,a) - \boldsymbol{\lambda}^\top \mathbf{C}(s,a)$). For the given $\boldsymbol{\lambda}$, if $a^*$ is an optimal action, then $\Pr\left(|\mathcal{Q}_{\boldsymbol{\lambda}}(h,a^*) - \mathcal{Q}_{\boldsymbol{\lambda}}(h,\hat{a})| > \nu\left[\sqrt{\frac{\log N(h,a^*)}{N(h,a^*)}} + \sqrt{\frac{\log N(h,\hat{a})}{N(h,\hat{a})}}\right]\right) \to 0$ as $t \to \infty$. In other words, all of the optimal actions are accepted asymptotically by the proposed equality test criterion.*

*Proof.* Without loss of generality, assume that $\mathcal{Q}_{\boldsymbol{\lambda}}(h,a^*) \geq \mathcal{Q}_{\boldsymbol{\lambda}}(h,\hat{a})$ at time $t$. Then, we can derive the following inequality, similarly to Lemma 3:

$$\Pr\left(\mathcal{Q}_{\boldsymbol{\lambda}}(h,a^*) - \mathcal{Q}_{\boldsymbol{\lambda}}(h,\hat{a}) > \nu\left[\sqrt{\frac{\log N(h,a^*)}{N(h,a^*)}} + \sqrt{\frac{\log N(h,\hat{a})}{N(h,\hat{a})}}\right]\right)$$

$$\leq \Pr\left(\mathcal{Q}_{\boldsymbol{\lambda}}(h,a^*) - \mathcal{Q}^*_{\boldsymbol{\lambda}}(h,a^*) > \nu\sqrt{\frac{\log N(h,a^*)}{N(h,a^*)}}\right)$$

$$+ \Pr\left(\mathcal{Q}^*_{\boldsymbol{\lambda}}(h,\hat{a}) - \mathcal{Q}_{\boldsymbol{\lambda}}(h,\hat{a}) > \nu\sqrt{\frac{\log N(h,\hat{a})}{N(h,\hat{a})}}\right) \tag{15}$$

where $\mathcal{Q}^*_{\boldsymbol{\lambda}} = \mathcal{Q}^{\pi^*_{\boldsymbol{\lambda}}}_{\boldsymbol{\lambda}}$. Note that $\mathcal{Q}^*_{\boldsymbol{\lambda}}(h,a^*) = \mathcal{Q}^*_{\boldsymbol{\lambda}}(h,\hat{a})$ since the both actions $a^*$ and $\hat{a}$ are optimal. It is sufficient to show that both of two terms in the right-hand side of Eq. (15) converge to 0 as $t \to \infty$. For any $M > 0$, the inequality

$$\nu\sqrt{\frac{\log N(h,a^*)}{N(h,a^*)}} \geq M \frac{\log N(h,a^*)}{N(h,a^*)}$$

holds as $N(h,a^*) \to \infty$. Therefore, by Eq. (13), the first term converges to probability 0. The second term also converges to probability 0 by the same reasoning. $\square$

Finally, we are now ready to provide the proof of Theorem 3:

*Proof of Theorem 3.* Let $B$ the event that the proposed equality test accepts all of the optimal actions while rejecting all of the suboptimal actions. Let $a^* = \arg\max_a \mathcal{Q}_{\boldsymbol{\lambda}}^{\oplus}(h, a)$. Then, for $B$ not to be satisfied, one of the following three cases should be hold:

1. $B_1$: $a^*$ is not optimal action (i.e. $a^* \notin \arg\max Q_{\boldsymbol{\lambda}}^{\pi_{\boldsymbol{\lambda}}^*}(h, a)$) with respect to the scalarized reward $R - \boldsymbol{\lambda}^\top \mathbf{C}$.

2. $B_2$: $\hat{a}$ is a suboptimal action but it is accepted as an equally optimal action while $a^*$ is an optimal action.

3. $B_3$: $\hat{a}$ is an optimal action but it is rejected while $a^*$ is an optimal action.

Then,

$$\Pr(B) \geq 1 - \Pr(B_1) - \Pr(\neg B_1 \cap B_2) - \Pr(\neg B_1 \cap B_3).$$

Here, $\Pr(B_1)$, $\Pr(B_2)$, and $\Pr(B_3)$ converge to zero as $t \to \infty$ by Lemma 2, 3, and 4 respectively, which concludes the proof. $\qquad\square$

## Appendix F  Proof of Theorem 2

To facilitate formal analysis, we assume that SEARCH of CC-POMCP is given by Algorithm 2. We first introduce the following lemma.

**Lemma 5.** *For any real numbers $V^*, V$, and $c > 0$, if there is $M > 0$ such that $|V^* - V| \leq M$ and $|V^* - c| > M$, then $(V^* - c)(V - c) > 0$.*

*Proof.*

$$
\begin{aligned}
(V^* - c)(V - c) &= (V^* - c)^2 + (V^* - c)(V - V^*) \\
&\geq (V^* - c)^2 - |V^* - c| \cdot |V - V^*| \\
&= |V^* - c|\big(|V^* - c| - |V - V^*|\big) \\
&> |V^* - c|\,(M - M) \\
&= 0. \qquad\qquad\qquad\qquad\qquad\qquad \square
\end{aligned}
$$

Now, we are ready to provide the proof of the following theorem, which guarantees that $\boldsymbol{\lambda}$ is improved until it converges to the optimal solution of Eq. (3), $\boldsymbol{\lambda}^*$.

**Theorem 2.** *Suppose $\boldsymbol{\lambda}$ is updated with increasing simulation step t, and the search tree is reset at the end of $\boldsymbol{\lambda}$'s update as outlined in Algorithm 2. If the asymptotic bias of UCT holds for all types of cost values (i.e. $\exists M > 0, \forall k, |V_{C_k}^{\pi_{\boldsymbol{\lambda}}^*}(h_0) - V_{C_k}(h_0)| \leq M\frac{\log t}{t}$), then either $sign(V_{C_k}^{\pi_{\boldsymbol{\lambda}}^*}(h_0) - \hat{c}_k) = sign(V_{C_k}(h_0) - \hat{c}_k)$ or $|V_{C_k}^{\pi_{\boldsymbol{\lambda}}^*}(h_0) - \hat{c}_k| \leq M\frac{\log t}{t}$ holds with probability 1 as $t \to \infty$, where $V_{C_k}(h_0)$ is an averaged Monte-Carlo return for k-th cost at the root node $h_0$.*

*Proof.* If $|V_{C_k}^{\pi_{\boldsymbol{\lambda}}^*}(h_0) - \hat{c}_k| \leq M\frac{\log t}{t}$, the statement trivially holds. If $|V_{C_k}^{\pi_{\boldsymbol{\lambda}}^*}(h_0) - \hat{c}_k| > M\frac{\log t}{t}$, then by Lemma 5 and the assumption $|V_{C_k}^{\pi_{\boldsymbol{\lambda}}^*}(h_0) - V_{C_k}(h_0)| \leq M\frac{\log t}{t}$,

$$(V_{C_k}^{\pi_{\boldsymbol{\lambda}}^*}(h_0) - \hat{c}_k)(V_{C_k}(h_0) - \hat{c}_k) > 0$$

which concludes $sign(V_{C_k}^{\pi_{\boldsymbol{\lambda}}^*}(h_0) - \hat{c}_k) = sign(V_{C_k}(h_0) - \hat{c}_k)$. $\qquad\square$

Even though Theorem 2 asymptotically guarantees that $\boldsymbol{\lambda}$ is near-optimal or it is updated by the direction of the negative subgradient, it requires *resetting* the entire search tree as described in Algorithm 2, which significantly degrades the sample efficiency of the algorithm. Fortunately, we can further show that the policy ordering is locally preserved even when $\boldsymbol{\lambda}$ is changed slightly, which justifies the use of practical Algorithm 1 that does not reset the tree and accumulates experiences on different lambda into single search tree.

**Theorem 4.** *For any $\boldsymbol{\lambda}_0$, there exists $\epsilon > 0$ such that if $\|\boldsymbol{\lambda}_1 - \boldsymbol{\lambda}_0\|_1 < \epsilon$ then $\forall b, \mathcal{V}_{\boldsymbol{\lambda}_0}^{\pi_1}(b) > \mathcal{V}_{\boldsymbol{\lambda}_0}^{\pi_0}(b)$ $\Rightarrow \forall b, \mathcal{V}_{\boldsymbol{\lambda}_1}^{\pi_1}(b) > \mathcal{V}_{\boldsymbol{\lambda}_1}^{\pi_0}(b)$. In other words, for a sufficiently small change of $\boldsymbol{\lambda}$, the ordering of the policies is preserved.*

*Proof.* Let $\Delta\boldsymbol{\lambda} = \boldsymbol{\lambda}_1 - \boldsymbol{\lambda}_0$. Then, for any $b$ and $\pi$,

$$
\begin{aligned}
\mathcal{V}_{\boldsymbol{\lambda}_1}^{\pi}(b) &= V_R^{\pi}(b) - \boldsymbol{\lambda}_1^{\top} V_{\mathbf{C}}^{\pi}(b) && \text{(Lemma 1)} \\
&= V_R^{\pi}(b) - (\boldsymbol{\lambda}_0 + \Delta\boldsymbol{\lambda})^{\top} V_{\mathbf{C}}^{\pi}(b) \\
&= \mathcal{V}_{\boldsymbol{\lambda}_0}^{\pi}(b) - \Delta\boldsymbol{\lambda}^{\top} V_{\mathbf{C}}^{\pi}(b)
\end{aligned}
$$

Now assume $\forall b, \mathcal{V}_{\boldsymbol{\lambda}_0}^{\pi_1}(b) > \mathcal{V}_{\boldsymbol{\lambda}_0}^{\pi_0}(b)$, let $\epsilon = \min_b \left[ \frac{1-\gamma}{C_{\max}} (\mathcal{V}_{\boldsymbol{\lambda}_0}^{\pi_1}(b) - \mathcal{V}_{\boldsymbol{\lambda}_0}^{\pi_0}(b)) \right] > 0$, and suppose $\|\Delta\boldsymbol{\lambda}\|_1 = \|\boldsymbol{\lambda}_1 - \boldsymbol{\lambda}_0\|_1 < \epsilon$. Then, for any $b$,

$$
\begin{aligned}
\mathcal{V}_{\boldsymbol{\lambda}_1}^{\pi_1}(b) - \mathcal{V}_{\boldsymbol{\lambda}_1}^{\pi_0}(b) &= \mathcal{V}_{\boldsymbol{\lambda}_0}^{\pi_1}(b) - \mathcal{V}_{\boldsymbol{\lambda}_0}^{\pi_0}(b) - \Delta\boldsymbol{\lambda}^{\top}(V_{\mathbf{C}}^{\pi_1}(b) - V_{\mathbf{C}}^{\pi_0}(b)) \\
&\geq \mathcal{V}_{\boldsymbol{\lambda}_0}^{\pi_1}(b) - \mathcal{V}_{\boldsymbol{\lambda}_0}^{\pi_0}(b) - \|\Delta\boldsymbol{\lambda}\|_1 \|V_{\mathbf{C}}^{\pi_1}(b) - V_{\mathbf{C}}^{\pi_0}(b)\|_{\infty} && \text{(Hölder's inequality)} \\
&> \mathcal{V}_{\boldsymbol{\lambda}_0}^{\pi_1}(b) - \mathcal{V}_{\boldsymbol{\lambda}_0}^{\pi_0}(b) - \epsilon \frac{C_{\max}}{1-\gamma} \\
&= \mathcal{V}_{\boldsymbol{\lambda}_0}^{\pi_1}(b) - \mathcal{V}_{\boldsymbol{\lambda}_0}^{\pi_0}(b) - \min_{b'} \left[ (\mathcal{V}_{\boldsymbol{\lambda}_0}^{\pi_1}(b') - \mathcal{V}_{\boldsymbol{\lambda}_0}^{\pi_0}(b')) \right] \geq 0
\end{aligned}
$$

$\square$

## Appendix G    Analytic Solution of LP (10) (when $K = 1$)

When $K = 1$ and $\lambda > 0$, LP (10) can be rewritten as:

$$
\begin{aligned}
\min_{\{w_i, \xi^+, \xi^-\}} \quad & (\xi^+ + \xi^-) \\
s.t. \quad & \sum_{i:a_i^* \in A^*} w_i Q_C(h, a_i^*) = \hat{c} + (\xi^+ - \xi^-) \\
& \sum_{i:a_i^* \in A^*} w_i = 1 \ \text{ and } \ w_i, \xi^+, \xi^- \geq 0
\end{aligned}
$$

Let $a_{\min} = \arg\min_{a_i^* \in A^*} Q_C(h, a_i^*)$ and $a_{\max} = \arg\max_{a_i^* \in A^*} Q_C(h, a_i^*)$. Then, the analytic solution is given by:

$$
\pi(a_{\min}|h) = \begin{cases} 0 & \text{if } Q_C(h, a_{\max}) \leq \hat{c} \\ 1 & \text{if } Q_C(h, a_{\min}) \geq \hat{c} \\ \frac{Q_C(h, a_{\max}) - \hat{c}}{Q_C(h, a_{\max}) - Q_C(h, a_{\min})} & \text{if } Q_C(h, a_{\min}) < \hat{c} < Q_C(h, a_{\max}) \end{cases}
$$

and

$$
\pi(a_{\max}|h) = \begin{cases} 1 & \text{if } Q_C(h, a_{\max}) \leq \hat{c} \\ 0 & \text{if } Q_C(h, a_{\min}) \geq \hat{c} \\ \frac{\hat{c} - Q_C(h, a_{\min})}{Q_C(h, a_{\max}) - Q_C(h, a_{\min})} & \text{if } Q_C(h, a_{\min}) < \hat{c} < Q_C(h, a_{\max}) \end{cases}
$$

This has only $O(|A|)$ time complexity, which is identical to that of UCB1 action selection.

## Appendix H    Experimental Setup

We used the following experimental parameters for each domain.

| Domain | Toy | Rocksample | Atari Pong |
|---|---|---|---|
| $\kappa$ | 1 | 20 | 0.1 |
| $\tau$ | $\hat{c} = 0.95$ | $\hat{c} = 1$ | $\hat{c} \in \{20, 30, 50, 100, 200\}$ |
| $\alpha_n$ | $1/n$ | $1/n$ | $1/n$ |
| $\nu$ | 1 | 1 | 1 |
| # of runs | 1000 | 100 (*) | 40 |
| # of simulations | from $2^3$ to $2^{20}$ | from $2^3$ to $2^{20}$ | 1000 |
| maximum-depth $d$ | $\gamma^d = 0.001$ | $\gamma^d = 0.001$ | $d = 100$ |

Table 2: (*) We report the result averaged over 100 runs or 12 hours of total computation time.

## Appendix I    Experimental results on Rocksample (15, 15)

Note that the baseline agent (red) is violating the cost constraint, so its return is not meaningful at all.