[Reviews · NeurIPS 2018]

Reviewer 1



This paper addresses a potentially important problem by giving an algorithm that can solve large constrained POMDPs with online methods. A constrained POMDP, which augments a traditional POMDP with multi-attribute cost constraints, is an important extension that can help model a wider range of real-world phenomena than a POMDP can. Having such an algorithm for solving large CPOMDPs is a very valuable contribution. The authors provide, in this paper, a derivation of an unconstrained objective to be solved (resulting from taking the dual of the CPOMDP's linear program), backed by theoretical justification, and an adaptation of the online search algorithm, POMCP, that incorporates cost constraints by approximately optimizing the objective. The paper is extremely well-written, free of typos, and clear in its presentation. I especially liked the separation between the derivation of the objective being minimized (Section 3: eq. 3 and 4, leading up to the three-step process outlined at the end) and the approximation algorithm CC-POMCP (Section 4). I noticed there was no related work section, except for the paragraph starting "Although the CPOMDP..." in the Introduction, but maybe this is okay due to the minimal previous work on this subject. (I don't know of any obviously missing references.) I am assuming that the derivation, starting from section 3, is your novel work (the theorem 1 procedure is very cool). If that is not true, then please indicate more clearly what aspects of the paper are novel. I was somewhat surprised by the result that the optimal policy for a CPOMDP is generally stochastic. It would be great if you could give a small example illustrating that fact, just for general edification of the reader. Several experiments are conducted to show the effectiveness of the proposed algorithm. They are well done, but ultimately somewhat disappointing. - The Rocksample domains, even though they are widely used to benchmark POMDP solvers, are actually really surprisingly easy to solve. The reason is that the uncertainty about the individual rocks is basically completely uncoupled, and so it's really more of a large collection of small POMDPs with something like a TSP to connect them. Further, the small POMDPs are relatively easy, given the actual observation probabilities, and there's very little branching in the optimal policy. - Given the nice ability of CPOMDPs to handle MA problems, it would have been good to see a domain that illustrates that. I could imagine something like oil exploration where: information about one test well actually give information about others (unlike the rocks) and where there are budgets for number of wells, distance traveled, total energy expended, etc. - The Pong domain seems very contrived, and seems to me to do a disservice to a very beautiful algorithm and paper. When describing your main algorithm CC-POMDP, I think it would be very helpful to readers if you were to identify the differences from the standard POMCP algorithm. For example, perhaps you could use a different color to indicate those differences. --------------- After reading the author response and other reviews, my opinion above still holds.

Reviewer 2



The paper proposes a online method for solving constrained POMDPs (CPOMDPs) using Monte-Carlo tree search (MCTS). The proposed foundation of the algorithm uses the dual of the belief state MDP with a scalarized reward over the reward function and scaled non-negative cost (with parameters lambda). Its key insight is that if parameters lambda are fixed, and are valid to satisfy the constraints, then it is simply a normal unconstrained POMDP. Thus, the procedure solves a belief MDP with fixed lambda, evaluates the costs, and intelligently improves parameters to ensure they do not violate the cost constraints. On top of this, a cost-constrained POMCP algorithm is used instead to approximately solve the unconstrained POMDP at each step. A theoretical analysis shows the PWLC of this scalarized belief state MDP, that the negative subgradient improves the objective, and that the algorithm converges in the limit (under some assumptions). Experiments compared a naive baseline agent (ensures a local action does not violate constraint, if so a random action is taken), CALP, and the proposed CC-POMCP on toy domain and three RockSample domains. Additionally, to show its scalability, results for the Pong CMDP successfully show the effect of increasing the constraints that the agent remain near the center. The paper provides a decent introduction to CPOMDPs, including a brief motivation and survey of related methods. While it is not surprising that MCTS performs well on these larger problems, the formulation of the constrained belief MDP and solution to handle the parameters (lambda) is simple with good mathematical support. There was one concern, however, regarding the guarantee that lambda always produce a feasible policy: 1. Proposition 2, which as cited comes from Lemma 4 in [13], requires that the rewards be non-negative (as well as the other assumption). Only under this assumption, does it make sense to clip lambda to [0, Rmax / (tau * (1 - gamma))]. This important assumption was not properly emphasized in the paper, as many (C)POMDPs do not have this property. If needed, then it would limit the application of the proposed algorithm. Is there any way to fix this limitation? Perhaps it is not a concern, if so, why? As it is an essential component that ensures the constraints are not violated, is there perhaps another way to ensure they are not and still use POMCP? Overall, the paper clearly describes the problems surrounding solving CPOMDPs and presents a solution with sufficient convincing evidence that it works in practice. While there is a concern raised above, the paper still contributes a good solution for solving CPOMDPs online with another MCTS-based approach.

Reviewer 3



This paper present CC-POMCP, an online MCTS algorithm for large CPOMDP that leverages the optimization of LP-induced parameters and only requires a black-box simulator of the environment. CPOMDP is a generalization of POMDP for multi-objective problems. Similar to POMDP, it can be cast into an equivalent belief-state CMDP and solved by a LP (Eq. 1). Given the dual of this LP (Eq. 2), the problem becomes an unconstrained belief state MDP with the scalarized reward function with the unknown weights. Then the basic idea of the CC-POMCP algorithm is to iterate through the steps of solving the belief-state MDP and updating the scalar weights with the POMCP method. The CC-POMCP algorithm is evaluated on the three benchmark problems, i.e., Toy, Rocksample, and Atari 2600 PONG. The LP and its dual for solving CMDP and POMCP for solving POMDP are not new but using POMCP and the dual for solving CPOMDP is novel. The main weakness of this work is that it does not fit the CPOMDP scenarios. The motivation of modeling a problem as a CPOMDP is that the constraints are hard constraints and must not be violated during execution. For UAVs under search and rescue mission, the main goal would be to find as many targets as possible (modeled as the reward function) while avoiding running out of battery (modeled as the constraints). Generally, such problems cannot be modeled as an unconstrained POMDP with the scalarized reward function. Based on the strong duality theorem, it seems that POMDP with constraints is equivalent to POMDP with the scalarized reward function. This is only true if the optimal policy is found. Unfortunately, POMCP can only found the approximate policy with limited time and resources. In such cases, there is no guarantee that the constraints are not violated. Without such property, the proposed algorithm is not useful for domains that must be modeled as a CPOMDP.